# Interaction between Macrophages and Adipose Stromal Cells Increases the Angiogenic and Proliferative Potential of Pregnancy-Associated Breast Cancers

**DOI:** 10.3390/cancers15184500

**Published:** 2023-09-10

**Authors:** Michael Doyle, Noor Kwami, Jaitri Joshi, Lisa M. Arendt, Jessica McCready

**Affiliations:** 1Department of Biological and Physical Sciences, Assumption University, 500 Salisbury St., Worcester, MA 01609, USA; 2Department of Comparative Biosciences, University of Wisconsin-Madison, 2015 Linden Drive, Madison, WI 53706, USA

**Keywords:** adipose stromal cells, pregnancy associated breast cancer, macrophages, co-culture

## Abstract

**Simple Summary:**

Breast cancer diagnosed during pregnancy and lactation (pregnancy-associated breast cancer, PABC) is more aggressive and has a decreased survival rate compared to breast cancers diagnosed at other stages in life. Previous research indicates that developmental changes to the fat cells in the mammary gland in lactation increase blood vessel growth within tumors. The aim of this report was to determine if other cells within the tumor contribute to PABC aggressiveness. Data presented here indicate that tumor cells transplanted with fat cells isolated during lactation increase inflammatory cells called macrophages and promote tumor cell growth in the mammary glands of mice. The growth of fat cells and macrophages together in a culture increases production of factors that enhance their ability to form blood vessels. These findings provide explanations for the increased aggressiveness of PABCs and suggest new treatment avenues.

**Abstract:**

Pregnancy associated breast cancers (PABCs) exhibit increased aggressiveness and overall poorer survival. During lactation, changes take place in the breast tissue microenvironment that lead to increased macrophage recruitment and alterations in adipose stromal cells (ASC-Ls). The interaction of these cells in PABCs could play a role in the increased aggressiveness of these cancers. We utilized an in vitro co-culture model to recreate the interactions of ASC-Ls and macrophages in vivo. We performed qRT-PCR to observe changes in gene expression and cytokine arrays to identify transcriptional changes that result in an altered microenvironment. Additionally, functional assays were performed to further elicit how these changes affect tumorigenesis. The co-culture of ASC-Ls and macrophages altered both mRNA expression and cytokine secretion in a tumor promoting manner. Tumorigenic cytokines, such as IL-6, CXCL1, CXCL5, and MMP-9 secretion levels, were enhanced in the co-culture. Additionally, conditioned media from the co-culture elevated the tumor cell proliferation and angiogenic potential of endothelial cells. These finds indicate that the changes seen in the microenvironment of PABC, specifically the secretion of cytokines, play a role in the increased tumorigenesis of PABCs by altering the microenvironment to become more favorable to tumor progression.

## 1. Introduction

Breast cancers diagnosed at different stages of a woman’s life vary in aggressiveness and prognosis. It has been observed that breast cancers diagnosed during pregnancy, lactation, and one year postpartum have an increased aggressiveness compared to breast cancers diagnosed outside of this time frame [1]. They have been called pregnancy-associated breast cancers or PABCs.

Breast tissue undergoes significant changes during and after pregnancy to accomplish the end goal of milk production. During pregnancy, epithelial cells receive signals from hormones to proliferate rapidly. During lactation, these cells produce milk. When milk is no longer needed by the offspring, the breast tissue undergoes involution, in which the cells move towards the pre-pregnancy state [2]. It is not just the epithelial cells that undergo changes during pregnancy, however. Adipocytes begin to lose some of their lipid content as a result of milk production [3]. Other stromal components, such as blood vessels, lymphatics, and immune cells, undergo changes that help promote the production of milk as well [4]. For example, macrophages are important for increasing the branching of epithelial cells during pregnancy. They also promote epithelial cell apoptosis that occurs during involution [5]. These changes that occur during pregnancy and lactation within the stroma could be a potential cause of the poorer prognosis of women with PABCs. Much of the research until recently has focused on epithelial cells and their role in cancer; however, newer research has begun to investigate the role of stromal cells in PABCs.

Macrophages have been found to compose up to 50% of a tumor’s mass [6] and can secrete cytokines to illicit an immune response. Classically activated macrophages are responsible for an inflammatory response and secrete cytokines, such as interleukin (IL)-6, IL-2, and tumor necrosis factor alpha (TNFα). Alternatively, activated macrophages are responsible for extracellular matrix remodeling and wound healing. The alternatively activated M2c subclass is responsible for releasing immunosuppressive cytokines, such as IL-10 and matrix metalloproteinases (MMPs) [7]. Macrophages have been shown to enhance the progression of premalignant lesions [8] and promote an aggressive phenotype in tumor cells, leading to metastasis [9]. Macrophage influx into the mammary gland during glandular involution has been shown to promote PABC. Potential treatment modalities have focused on the role macrophages play in PABC [5,10].

The role of adipocytes has become increasingly important in the understanding of PABCs. Adipocytes undergo changes during pregnancy and lactation, likely due to increased metabolic demands. Adipocytes account for most of the breast tissue in their nulliparous and regressed states while they compose only a small fraction of breast tissue during lactation [3]. Adipose stromal cells present during lactation (ASC-Ls) are smaller in size, exhibit signs of membrane folding, and are unable to store lipids. They also demonstrate increased expression of vascular endothelial growth factor-D (VEGF-D) and IL-6 and increase tumor growth by promoting angiogenesis [4,11]. The presence of VEGF-D and IL-6 have been used to determine whether an adipocyte has converted to a carcinoma-associated fibroblast (CAF) [12]. ASC-Ls are capable of promoting tumor growth even when harvested from non-diseased tissue [3]. This indicates that ASC-Ls alone can contribute to the formation and proliferation of tumors in non-diseased tissue and that the stroma is an important factor in tumorigenesis. Additionally, ASCs express increased levels of inflammatory cytokines, which elicit increased macrophage recruitment, thereby amplifying the effects of macrophages [11,13].

The role of adipocytes in the breast tissue microenvironment is important in the aggressiveness of PABC. Macrophages are also important, if not critical to the proliferation of breast tumors; however, the relationship between macrophages and adipocytes is largely unknown. It is assumed that the cytokines released by each play a role in converting the other to become more CAF-like. There is a gap in in our understanding of the interaction of ASCs and macrophages; therefore, we investigated this interaction to test if together they contribute to the increased aggressiveness of PABCs. We performed co-culture experiments with macrophages and ASC-Ls to study the changes that occur when these cells are grown in close proximity and to observe if these paracrine interactions could potentially alter the breast microenvironment in vivo. We found that ASC-Ls recruit macrophages to breast tumors causing an increased proliferation of tumor cells, angiogenesis, and cytokine secretion.

## 2. Materials and Methods

Cell culture. ASC-L lines were isolated from the mammary glands of Bab/c mice [1]. RAW 264.7 macrophages were a kind gift from Dr. Aisling Dugan (Brown University, Providence, RI, USA). All cell lines were cultured at 37 °C in 5% CO_2_. HUVEC cells were grown in endothelial growth media containing fibroblast growth factor-basic (bFGF; Sigma, St. Louis, MO, USA) and 5% fetal bovine serum (FBS). All other cells were cultured in DMEM with 5% FBS and 1% antibiotic-antimycotic. For co-culture experiments, 200,000 low passage ASC-Ls from 3 separate mice were plated directly with 250,000 RAW 264.7 macrophages in 6-well plates in growth media. Six hours after plating, the growth media was removed, cells were washed with PBS, and serum-free DMEM was added. Two days after the original plating, the conditioned media was removed and stored at −80 °C. The cells were scraped and removed from the plates and stored at −80 °C.

Cytokine Array Analysis. Cytokine array was obtained from RayBio, Norcross , GA, USA (AAM-NEU-1-2) and executed following the manufacturer’s protocol. The membranes were imaged using a BioRad chemiluminescence imager. The data were analyzed using the software provided by RayBio (AAM-NEU-1-SW). The cytokines that demonstrated increased expression when cells were co-cultured compared to isolated growth were selected for further analysis and validated using qRT-PCR.

Immunohistochemistry. Paraffin embedded sections of tumors generated in a previous study [3] were rehydrated in graded alcohol and underwent antigen retrieval in 0.01 M citrate buffer (pH = 6.0) at 95 °C for 20 min. Sections were blocked for 1 h in 5% bovine serum albumin (BSA) in PBS and incubated with anti-mouse F4/80 (1:250, eBiosciences, Waltham, MA, USA, 14-4801) or Ki67 (1:250, Abcam, Cambridge, UK, ab16667) at 4 °C overnight. For immunofluorescence, tissue sections of tumors generated in a previous study [3] were then incubated with Alexa Fluor 488 goat anti-rat IgG (Life Technologies, Carlsbad, CA, USA, A11006) diluted 1:250 in 5% BSA in PBS for 30 min. Sections were counter stained with 250 ng/mL DAPI for 5 min and mounted with the Slow-Fade mounting kit (Life Technologies, S2828). For immunohistochemistry, tissue sections of tumors generated in a previous study [3] were incubated with goat anti-rabbit IgG (Vector Laboratories, Newark, CA, USA, BA-1000) diluted 1:500 in 5% BSA in PBS for 30 min. Tissue was incubated with Vectastain ABC-HRP Kit (Vector Laboratories, PK-4000), followed by ImmPACT DAB substrate (Vector Laboratories, SK-4105) according to manufacturer’s instructions. Sections were counterstained with hematoxylin. Tumor tissue was imaged using fluorescent microscopy with NIS Elements BR 3.2 imaging software. Five images were quantified from five mice/group. All cell counting and image alignment was performed using Image J software.

Macrophage Migration Assays. An amount of 5 × 10^5^ RAW264.7 cells were plated in serum-free DMEM in transwell inserts containing 8 µm pores (Corning, Corning, NY, USA, 353097). Transwells were placed in serum-free media or conditioned media isolated from CommaD cells, ASC-L lines isolated from 3 different mice, and cultures of CommaD cells and each line of ASC-L. RAW264.7 cells migrated for 4 hrs at 37 °C; then, transwells were stained with crystal violet. Inserts were imaged using ISOcapture software. Three experiments were completed with all conditions plated in triplicate.

Functional Assays. Proliferation. An amount of 12,500 CommaD, 4T1-12B or HUVEC cells were plated in a 24-well plate in growth media. Six hours after plating, the growth media was removed, and the cells were washed with PBS and diluted (1-part conditioned media to 1-part DMEM). Conditioned media was added from one of three different conditions: RAW264.7 macrophages, ASC-Ls, or RAW264.7 and ASC-Ls cultured together (co-culture). Cells were counted using a hemocytometer on 1, 3, and 6 days after plating. Wound healing. An amount of 225,000 HUVEC cells were plated in 12-well plates in growth media. Six hours after plating, growth media was aspirated, and cells were starved with 2% FBS. Twelve hours after serum starvation, cells were washed with PBS, wounded with the tip of a p200 pipette tip, and conditioned media was added. Six hours after wounding, cells were imaged, and the percentage wound closure was determined. Tube formation. An amount of 0.5 mL Matrigel was added to each well in a 96-well plate and spun for 30 s at 21,000× *g*. The plate was incubated at 37 °C for 30 min while the cells were counted and prepared. An amount of 150,000 HUVEC cells were added to 500 μL of conditioned media. An amount of 100 μL of each sample was plated onto the Matrigel in duplicate and incubated at 37 °C. Four hours after plating, cells were imaged, and number of tubes formed were counted.

qRT-PCR. Collected cell pellets were lysed, and the mRNA was isolated and purified using Qiagen RNeasy kit following manufacturer’s protocol. cDNA was created using iScript reverse transcriptase and stored at −80 °C. qRT-PCR was performed using iTaq SYBR Green polymerase on cDNA samples in a thermocycler. Data were analyzed using a ΔΔCT analysis [14], and fold increases were compared to macrophages alone. Primer sequences were as follows: Actin F 5′ GACAGGATGCAGAAGGAGATCAC-3′ Actin R 5′ TCAGGAGGAGCAATGATCTTGA-3′ COX2 F 5′-TTCAACACACTCTATCACTGGC COX2 R 5′ AGAAGCGTTTGCGGTACTCAT CXCL5 F 5′-TGCGTTGTGTTTG- CTTAACCG-3′ CXCL5 R 5′-CTTCCACCGTAGGGCACTG-3′ IL-6 F 5′-ACAAAGCC-AGAGTCCTTCAGAG-3′ IL-6 R 5′-GTGAGGAATGTCCACAAACTGA-3′ MMP-9 F 5′-CTGGACAGCCAGACACTAAAG-3′ MMP-9 R 5′-CTCGCGGCAAGTCTTCAGAG-3′.

Statistical Analyses. Significance was determined at *p*-values of 0.05 or less. Data were tested for normality using the Shapiro–Wilk test prior to further statistical analysis. Statistical tests used are listed in the figure legends. Error bars represent mean ± SEM unless stated. Statistical analyses were conducted using GraphPad Prism 6.0 (GraphPad Software, San Diego, CA, USA).

## 3. Results

### 3.1. ASC-Ls Recruit Macrophages into the Tumor Microenvironment

Macrophages are present at higher levels in breast tissue during mammary gland remodeling, specifically during pregnancy and lactation [15]; therefore, macrophages and adipocytes intermingle in vivo. To determine whether this relationship is recreated in the breast tumor microenvironment, we analyzed tumors isolated from mice injected with the 4T1-12B tumor cell line [16] that were generated in a previous study [3]. These cells, when injected into the mammary fat pad of mice at each stage of development, recreate the pattern seen during PABC, with the largest tumors forming during lactation [3]. We performed immunofluorescence to quantify the number of macrophages in tumors isolated from mice co-injected with 4T1-12B mouse tumor cells and adipocytes isolated from nulliparous mice (ASC-N) and compared these tumors to those isolated from mice co-injected with 4T1-12B tumor cells and ASC-Ls (Figure 1a,b). Tumors co-injected with ASC-Ls have a significantly higher number of macrophages (*p* = 0.0025) indicating that macrophages move towards ASC-Ls in vivo (Figure 1c). To determine if ASC-Ls or tumor cells recruit macrophages into the tumor, we performed a migration assay with the breast cell line, CommaD cells (Figure 1d). These cells are closer to normal breast tissue but can progress towards tumors in the proper environment [17], thereby making them an ideal cell line for in vitro conditioned media experiments. When exposed to conditioned media from CommaD cells, ASC-Ls, RAW 264.7 macrophages, or a combination of the two increased migration towards the media isolated from all cell types when compared to serum-free media, but only showed a significant increase in movement when exposed to media isolated from ASC-L and Comma D cells combined (*p* = 0.03). This conditioned media is similar to the microenvironment of a tumor but isolates the contribution of the ASC-Ls without other stromal cell types present. These data indicate that ASC-Ls are capable of recruiting macrophages towards the tumor microenvironment of two different types of breast cancer cells: 4T1-12B in vivo and CommaD in vitro.

### 3.2. ASC-Ls and Macrophages Increase Breast Cancer Cell Proliferation

The recruitment of macrophages by ASC-Ls suggested that these cells may be interacting and affecting the behavior of tumor cells. To determine if breast cancer cells are affected when macrophages and ASC-Ls are in close proximity, such as in a tumor, we measured the cellular proliferation of CommaD cells when injected alone (Figure 2a) or co-injected with ASC-Ls (Figure 2b). Tumors isolated from mice co-injected with ASC-Ls proliferated significantly more than CommaD cells alone (*p* = 0.002, Figure 2c).

The in vivo data encouraged us to further investigate the interaction between macrophages and ASC-Ls in order to determine their effects on tumor cells. To this end, we used a co-culture model whereby macrophages and ASC-Ls were cultured in the same dish (Figure 2d), and their conditioned media was collected to use in a functional assay. There was an increase in proliferation of 4T1-12B cells cultured in conditioned media isolated from macrophages, ASC-Ls and co-culture when compared to cells receiving serum free media (Figure 2e). There was no significant difference when comparing the individual conditioned media. Taken together, these data indicate that macrophages and ASC-Ls increase the proliferation of breast cancer cells in vivo and in vitro.

### 3.3. Co-Culture of ASC-Ls and Macrophages Increases Angiogenesis

Endothelial cells are an important part of the tumor microenvironment and essential for the process of angiogenesis, a hallmark of cancer. Previously published research indicates that ASC-Ls increase tumor angiogenesis [3], leading us to investigate if macrophages were involved in this process. We performed an endothelial cell tube forming assay (Figure 3a) using HUVEC cells treated with conditioned media to determine the effect of conditioned media from macrophages and ASC-Ls on angiogenic potential. Conditioned media isolated from co-culture of macrophages and ASC-Ls significantly increased tube formation compared to conditioned media isolated from macrophages (*p* = 0.009) and ASC-Ls cultured alone (*p* = 0.04). We next performed proliferation assays to determine if HUVEC cells proliferated at a higher rate when exposed to conditioned media from co-cultured cells (Figure 3b). Similar to the breast cancer cell proliferation assay (Figure 2e), conditioned media from all cell types increased HUVEC cell proliferation. We also performed a wound healing assay to determine if HUVEC cells migrated faster when exposed to conditioned media isolated from co-culture cells (Figure 3c). Wound closure of HUVEC cells was not significantly faster when treated with co-culture media than when compared to movement of HUVEC cells in the presence of media isolated from macrophages or ASC-Ls alone. These data suggest that the co-culture of macrophages and ASC-Ls increases the ability of endothelial cells to create new blood vessels but does not increase their proliferation or migration over the levels that macrophages and ASC-Ls cause individually.

### 3.4. The Interaction of ASC-Ls and Macrophages Increases Cytokine Secretion

Our data show that ASC-Ls and macrophages increase tumor cell proliferation and endothelial cell tube formation. To begin to understand the mechanism of how these cells are changing the behavior of cells in the tumor microenvironment, we performed a cytokine array (Figure 4a) to measure the cytokines released when ASC-Ls and macrophages were cultured in the same dish. We identified four cytokines that were secreted at a higher amount in the conditioned media of co-cultured cells when compared to conditioned media of either macrophages alone or ASC-Ls alone: IL-6, CXCL1 (labeled as KC on the heat map), CXCL5 (labeled as LIX on heat map), and MCP-1. MIP-1alpha increased in the co-culture media when compared to ASC-L media alone, and MMP-3 increased in the co-culture media when compared to macrophage media alone. These are all cytokines that have been shown to increase tumor aggression [18,19,20,21,22,23,24].

To verify the results of the cytokine array, we performed qRT-PCR using RNA isolated from macrophages, ASC-Ls, and macrophages co-cultured with ASC-Ls (Figure 4b). RNA levels of IL-6 were significantly higher (*p* ≤ 0.03) in co-culture cells when compared to either macrophages or ASC-L cells cultured alone. CXCL5 RNA expression levels were significantly higher (*p* = 0.01) in co-culture cells when compared to macrophages grown in isolation. We also verified the levels of two genes not included in the cytokine array: MMP-9, a protein involved in angiogenesis [24,25] and COX-2, a protein previously implicated in PABC aggressiveness [26]. Co-culture of macrophages and ASC-Ls significantly increased expression of both MMP-9 (*p* = 0.01) as well as COX-2 (*p* = 0.03). These data indicate that crosstalk between macrophages and ASC-Ls promotes the expression of multiple cytokines associated with PABC.

## 4. Discussion

PABCs are diagnosed at a more advanced stage and have decreased mortality [1]; therefore, it is imperative to understand its pathogenesis. Our data show that ASC-Ls, isolated from normal mouse mammary glands [3], recruit macrophages to the tumor microenvironment, leading to increased tumor cell proliferation. The cross-talk seen during normal gland development may be re-created in the tumor microenvironment. Macrophages have been observed in close proximity to epithelial cells and stromal cells during lactation [5,27]. This suggests that our data presented here correctly mimics the environment seen in the developing gland and that these stromal cells are collaborating to increase tumor proliferation and angiogenesis, likely through the secretion of multiple cytokines.

The mammary gland is a complex organ that undergoes many changes throughout a woman’s lifetime. Cross-talk at various stages of development between stromal cells and the milk-producing epithelium causes functional changes in the gland. For example, during pregnancy, ASCs fill with lipids until lactation, at which time the ASCs undergo lipolysis to satisfy the energy needs of the lactating gland [3,28]. VEGF, a growth factor responsible for increased angiogenesis, is expressed by mammary epithelial cells while VEGF receptors are expressed by endothelial cells. This expression pattern changes throughout gland development [29]. Macrophages affect both epithelial cells and other stromal cells, such that they clear the gland of unneeded epithelial cells during post-lactational involution and allow for adipocyte repopulation [30]. Our data add to this growing list, suggesting that stromal cells are not simply present in the gland at the same time but that they play a critical role in communication with both the epithelium and immune cells.

Conditioned media from the stromal cells caused behavioral changes in two different breast cancer cell lines, the more aggressive 4T1-12B and the less aggressive CommaD cells. We also observed an increase in angiogenic potential of endothelial cells in response to conditioned media from macrophages and ASC-Ls. We propose that this is due to the secretion of multiple cytokines that have been shown to increase breast cancer aggressiveness, namely CXCL5, COX-2, MMP-9, and IL-6. Our data indicate that these cytokines were secreted at significantly higher amounts when ASC-Ls and macrophages were co-cultured. Multiple reports suggest that these cytokines increase tumor cell proliferation and promote angiogenesis [13,25,26,31,32,33]. Similar to our data, Zhao et al. [31] isolated conditioned media containing CXCL5 from adipose stem cells and observed increases in proliferation of both estrogen receptor-positive and -negative breast cancer cell lines. COX-2 has been a target for breast cancer treatment to decrease both tumor cell proliferation and angiogenesis [32] and has been identified as a target to improve PABC treatment [26]. MMP-9 is overexpressed in breast cancer [33] and cleaves components in the extracellular matrix to release VEGF [25], thereby increasing angiogenesis, among other tumorigenic roles. IL-6 has been shown to be secreted by cancer-associated adipocytes, leading to both increased angiogenesis and breast cancer proliferation [34]. Based on our data, we suggest that ASC-Ls behave similar to cancer-associated adipocytes (CAA) and drive proliferation and angiogenesis during PABC, leading to increased aggressiveness. The key characteristics of CAA are small dispersed lipid droplets and overexpression of adipokines including, but not limited to, IL-6. We have previously shown that ASC-Ls also have very small dispersed lipid droplets [3] and show here that ASC-Ls secrete pro-inflammatory cytokines, such as IL-6. The fact that ASC-Ls share the characteristics of CAAs prior to tumor formation may be one factor that promotes the aggressiveness of PABCs.

A limitation of our study is that we limited the collection of ASC-L used in our experiments to day 3 of lactation rather than exploring multiple time points during lactation. It is possible that the function of ASC-L changes over the course of lactation in response to hormones, such as prolactin and oxytocin, which are impacted by breastfeeding. It would be interesting to isolate ASC-Ls at various timepoints throughout lactation and determine if there is a difference in behavior when lactation is suppressed earlier versus later. However, if ASC-Ls are indeed behaving as CAAs and a pre-existing tumor was present in the gland, we hypothesize that ASC-Ls isolated at any timepoint would affect the aggressiveness of the tumor. While we have identified interactions that occur among macrophages and ASC-Ls, these studies focus on mouse tissue, and further studies are necessary to validate these interactions within lactating human breast tissue. However, significant challenges are present in obtaining the clinical samples required to address this research question.

The immunogenic capability of stromal cells of the lactating mammary gland have been underestimated in PABC. Analysis of the media isolated from co-cultured stromal cells shows that the cross-talk between ASC-Ls and macrophages increases the secretion of cytokines that promote tumor growth and angiogenesis. Iyengar et al. [35] and Reggiani et al. [36] have shown that the interaction of ASCs and macrophages in breast cancers of obese patients increases secretion of IL-6 and MMP-9, respectively, two cytokines we identified in our conditioned media. The increase in cytokine release in our experiments was not dependent on signaling from tumor cells, as our media were collected in the absence of any epithelial cells. These data suggest that macrophages and ASC-L may play a role in early tumor growth to generate a supportive environment for the rapid progression observed in PABC. It is also possible that the ASC-Ls and tumor cells cooperate to recruit the macrophages during tumor progression. Additional studies are warranted to understand this communication paradigm and determine whether the interruption of cytokine signaling caused by the interaction of ASC-Ls and macrophages affects tumor aggressiveness.

## 5. Conclusions

This study is the first of its kind to identify CXCL1, CXCL5, COX-2, IL-6, MMP-9, and MCP-1 as secreted from adipose stromal cells in the normal lactating mammary gland. Our data suggest that the interactions of macrophages and ASC-Ls in the lactating mammary gland drives tumor aggression. Interruption of the signaling between these cells may be a novel therapeutic option.

## Figures and Tables

**Figure 1 cancers-15-04500-f001:**
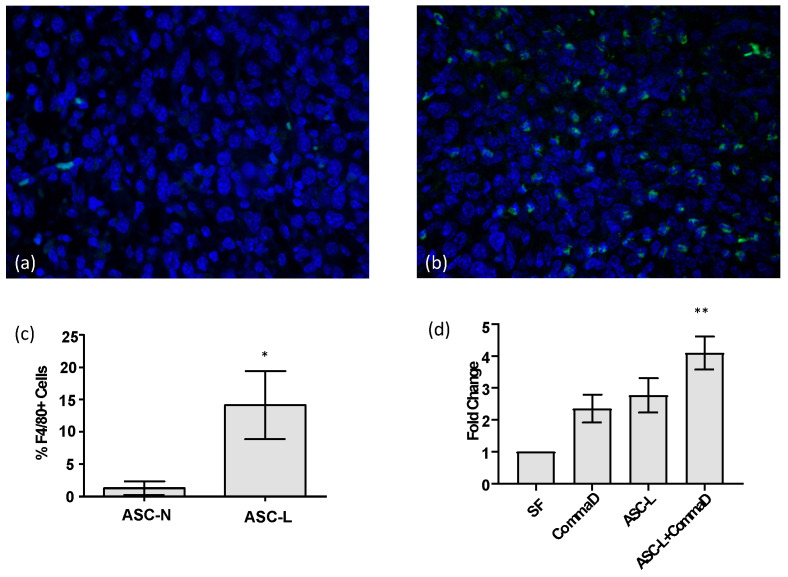
ASC-Ls recruit macrophages into the tumor microenvironment. (**a**) Representative immunofluorescence image of breast cancer tissue isolated from mice co-injected with 4T1-12B cells and ASC-Ns stained with F4/80+ (green) and DAPI (blue). (**b**) Representative immunofluorescence image of breast cancer tissue from mice co-injected with 4T1-12B cells and ASC-Ls stained with F4/80+ (green) and DAPI (blue). (**c**) Quantification of macrophages present in tumors co-injected with 4T1-12B cells and ASC-Ns or 4T1-12B cells co-injected with ASC-Ls (*n* = 6 tumors/group). Mann–Whitney U test; * *p* = 0.0025 (**d**) Quantification of migration assay measuring migration of RAW 264.7 macrophages towards serum-free media (SF) or conditioned media. Repeated measures ANOVA with Dunn’s multiple comparison test; SF versus ASC-L + CommaD ** *p* = 0.03.

**Figure 2 cancers-15-04500-f002:**
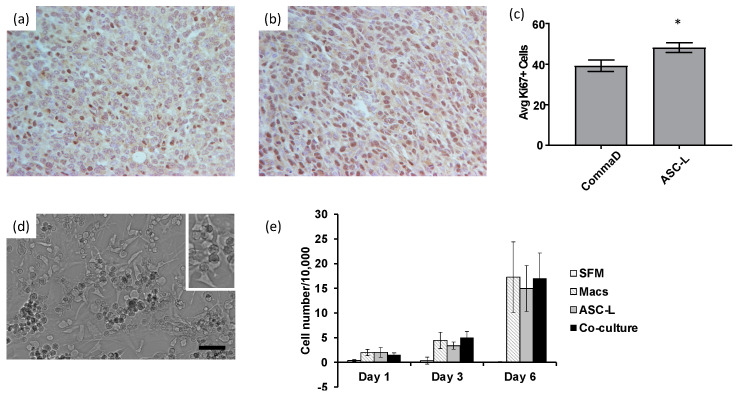
ASC-Ls and macrophages increase proliferation of breast cancer cells. (**a**) Representative immunohistochemistry image of breast cancer tissue isolated from mice injected with CommaD cells stained with Ki67. (**b**) Representative immunohistochemistry image of breast cancer tissue isolated from mice co-injected with CommaD cells and ASC-Ls stained with Ki67. (**c**) Quantification of Ki67 staining measuring proliferation of CommaD cells. * *p* = 0.02 Mann–Whitney U Test (*n* = 5 tumors/group). (**d**) Representative image of cells grown in co-culture. Macrophages are the more rounded, darker cells while ASC-Ls are lighter and more stellate. Scale bar = 50 μm. (**e**) Proliferation assay using conditioned media on 4T1-12B cells (*n* = 4).

**Figure 3 cancers-15-04500-f003:**
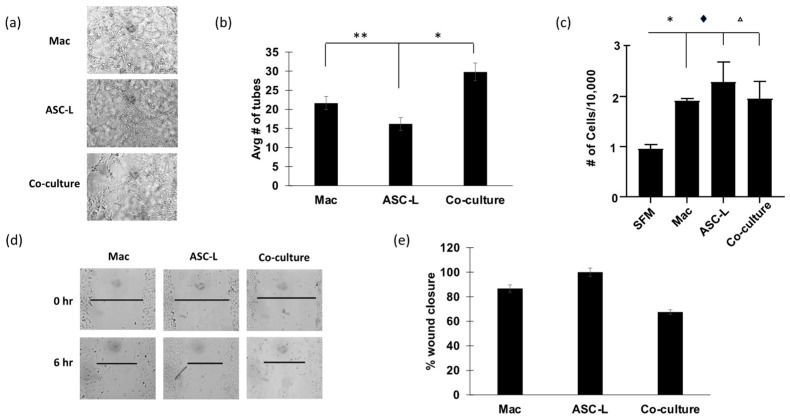
Co-culture increases angiogenic potential of HUVEC cells. (**a**) Representative images of tube forming assay using HUVEC cells. (**b**) Quantification of tube forming assay. * *p* = 0.04, ** *p* = 0.009 Student’s t-test (*n* = 3). (**c**) Proliferation assay day 3 using conditioned media on HUVEC cells. * *p* = 0.005, ♦ = 0.03, Δ = 0.04. One-way ANOVA with Tukey’s Multiple Comparison test. *n* = 3 (**d**) Representative images of wound healing assay using conditioned media on HUVEC cells. (**e**) Quantification of wound healing assay. Macrophage and co-culture wound closure relative to ASC-L, which was set to 100%. *n* = 3.

**Figure 4 cancers-15-04500-f004:**
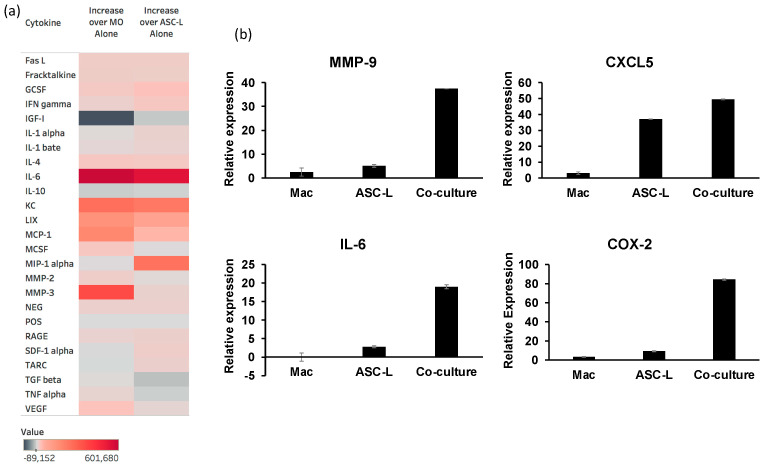
Molecular characterization of the interaction between ASC-Ls and macrophages. (**a**) Heat map of the quantification of cytokines from cytokine array compared to macrophages alone (MO) and ASC-Ls alone. (**b**) qRT-PCR verification of cytokine array data using primers for MMP9, CXCL5, IL-6, and COX-2. *n* = 3.

## Data Availability

Not applicable.

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
