# Peer review of "Interaction between Macrophages and Adipose Stromal Cells Increases the Angiogenic and Proliferative Potential of Pregnancy-Associated Breast Cancers"

_cancers, 2023, doi:10.3390/cancers15184500_

Round 1

Reviewer 1 Report

Dear Authors,

This article focuses on an important topic related to the potential clinical implications of the interaction of macrophages and adipose stromal cells in the cell signaling of PABC. The present study shows the usefulness of quantifying the role of macrophages and ASC-Ls in tumor aggressiveness and the possibility of their application in clinical practice. Since the literature has scarce data, this study is essential in identifying the mechanisms that will underlie the best experimental and clinical practices. The authors provided adequate details on methodology, evaluation, findings, and investigations. The particularities and novelty of the article are very well underlined in the results and conclusions sections. Given the bibliography, it is clear that the authors made a complete review of the literature beforehand. 

However, some suggestions could improve the quality of the article:

I would like the authors to identify and report some limitations of this study.

- How does the duration of lactation influence this interaction between macrophages and ASC-Ls regarding tumor aggressiveness? Would the early suppression of lactation improve the prognosis of these cases?

Kind regards

Author Response

We thank the reviewer for the thoughtful comments.   We have made the following changes to address your concerns:

- I would like the authors to identify and report some limitations of this study.

We have added information regarding the limitations of our study in the discussion section (lines 364-376).

- How does the duration of lactation influence this interaction between macrophages and ASC-Ls regarding tumor aggressiveness? Would the early suppression of lactation improve the prognosis of these cases?

 The ASC-Ls we used for all of our experiments were isolated from lactating mice on day 3, thereby essentially cutting lactation short of the typical 21 days.  We would hypothesize that if there is a pre-existing tumor in the gland during lactation, the duration of lactation may affect the aggressiveness of the tumor.  However, since we did not isolate ASC-Ls at various timepoints we can only speculate to this point.  We have added a section in the discussion to address this interesting idea as a future experiment (lines 368-372).

Reviewer 2 Report

In this manuscript by Doyle et al, the authors reported that macrophages and adipose tissue within the tumor microenvironment contribute for aggressive pregnancy associated breast cancer development. Using invitro co-culture models, they have reported that conditioned medium from cocultures potentiated tumor cell proliferation and angiogenic potential of endothelial cells. They further continue to show that co-culture of macrophages and adipocytes enhanced titers of cytokines that were known to induce tumor growth and progression.    Though the focus of the study was interesting in terms of understanding the tumor microenvironment, particularly in pregnancy associated breast cancers, the authors may consider few concerns that the authors might consider strengthening their work.

1.     In the results section, 3.1, it was written “To determine whether this relationship is recreated in the breast tumor microenvironment, we analyzed tumors isolated from mice injected with the 4T1-12B tumor cell line [16]. These cells, when injected into the mammary fat pad of mice at each stage of development, recreate the pattern seen during PABC, with the largest tumors forming during lactation [3].” It is not clear whether these experiments were done or details from published articles are being discussed !

2.     In Fig 1C, as the comparison is being made between ASC-N and ASC-Ls while the tumors cells are 4T1-12B, the x-axis need changes accordingly.

3.     In Fig 3, providing the images of endothelial cell tube formation with different cultural conditions will be more useful for understanding the strength of the observation.

4.     Though the authors have reported that macrophages and adipocytes interaction are complementing for tumor progression, what the contributing factors triggering the outcome because of this interaction and the details of the interaction are not convincing. A detailed analysis of this interaction may strengthen the authors observation.

5.     Do the authors observation reflect with early stages of pregnancy as well as during mammary gland involution?

The language was poorly presented. At times it was difficult to under the flow of the story, particularly while making a comparative study. 

Author Response

We thank the reviewer for the comments.  We have edited the revised manuscript and improved the background and references in the introduction. 

  1. In the results section, 3.1, it was written “To determine whether this relationship is recreated in the breast tumor microenvironment, we analyzed tumors isolated from mice injected with the 4T1-12B tumor cell line [16]. These cells, when injected into the mammary fat pad of mice at each stage of development, recreate the pattern seen during PABC, with the largest tumors forming during lactation [3].” It is not clear whether these experiments were done or details from published articles are being discussed !

We apologize for the confusion.  We have completed a new analysis using tumors generated in a previous study.  We have revised this statement to make the experimental details clear (line 188). 

  1. In Fig 1C, as the comparison is being made between ASC-N and ASC-Ls while the tumors cells are 4T1-12B, the x-axis need changes accordingly.

Thank you for pointing out this error.  We have edited the x-axis in Figure 1C to reflect the correct information.

  1. In Fig 3, providing the images of endothelial cell tube formation with different cultural conditions will be more useful for understanding the strength of the observation.

We agree that showing the images of the wound healing strengthens our data.  We have edited figure 3 to include representative images of both time 0 (the start) and 6 hours later (the end) of our wound healing experiments.

  1. Though the authors have reported that macrophages and adipocytes interaction are complementing for tumor progression, what the contributing factors triggering the outcome because of this interaction and the details of the interaction are not convincing. A detailed analysis of this interaction may strengthen the authors observation.

We agree with the reviewer’s comment that a thorough understanding of the interaction between macrophages and adipocytes will lead to a better understanding of tumor progression. We believe this is one reason our paper is important.  In Figure 4, we have identified cytokines secreted individually by macrophages and ASC-L, as well as factors that are enhanced when macrophages and adipocytes are cultured together.  Many of these factors have been shown to enhance tumor progression using both in vitro and in vivo model systems.  Likely, the mechanism of these interactions is complex as these cytokines and growth factors impact both the tumor cells as well as endothelial cells to enhance angiogenesis.  We have clarified our description of the potential interactions of these cytokines within the tumor microenvironment in the discussion section (lines 339-363).    

  1. Do the authors observation reflect with early stages of pregnancy as well as during mammary gland involution?

In our previously published paper (McCready et al 2014), we conducted a thorough characterization of adipocytes isolated from each stage of mammary gland development, including pregnancy and mammary gland involution.  We observed that adipocytes isolated during lactation caused a significantly greater increase in tumor growth and angiogenesis compared to adipocytes isolated during any other stage of mammary gland development.  For this reason, we focused on ASC-Ls in this manuscript. 

Reviewer 3 Report

Comment 1:

Fig 1(a), 1(c): Authors have measured number of macrophages in tumors isolated from mice by immunofluorescence. To look at the population, it should be cross validated using another approach like FACS where you could quantify whole population. 

Comment 2:

Fig 1(d): Were macrophages serum starved before performing the migration assay? Usually cells tend to move towards conditioned media if placed in a serum free medium (chemoattractant). There is an additive effect on macrophage migration upon giving combined conditioned media ASCL+CommaD. How much is migration when authors give conditioned media from  ASC-Ns? This will serve as a better control.

Comment 3:

Fig. 2 (e): Authors claim that there is increase proliferation of breast cancer cells in conditioned media co-cultured from macrophage+ASC-Ls. There are huge error bars (are these Standard deviations)? It is hard to interpret significance level for individual conditioned media vis-a-vis co-culture. What is being quantified is not clear here? Is this assay similar to colony formation assay? Please clarify. Authors should show a better representative image for Fig. 2 (d). There is no scale bar. 

Comment 4:

Conditioned media has various factors and in general known to promote proliferation. How can we attribute this effect to specifically macrophage proliferation? There should be some control to make this claim.

Comment 5:

Fig 3 (b): Increased cellular proliferation should be cross validated using FACS. Fig. 3 (c) To measure HUVEC cells migration why authors preferred wound healing assay over transwell migration? Any specific reason? Is 6 hr enough for HUVEC cells for wound closure ?

Author Response

We thank the reviewer for the thoughtful comments. 

  1. Fig 1(a), 1(c): Authors have measured number of macrophages in tumors isolated from mice by immunofluorescence. To look at the population, it should be cross validated using another approach like FACS where you could quantify whole population. 

For this experiment, our goal was to examine both total macrophages and their localization within the tumor.  We have used a macrophage marker which is inclusive of the majority of macrophage populations within a tumor.  Since these tumors were collected as a part of our previously published work (McCready et al 2014), we do not have samples from these tumors for flow cytometry analysis. 

  1. Fig 1(d): Were macrophages serum starved before performing the migration assay? Usually cells tend to move towards conditioned media if placed in a serum free medium (chemoattractant). There is an additive effect on macrophage migration upon giving combined conditioned media ASCL+CommaD. How much is migration when authors give conditioned media from ASC-Ns? This will serve as a better control.

Yes, the cells were serum starved before the migration assay.  The data are represented as a fold change compared to macrophage migration in the presence of serum-free media.  In our previous work, we characterized ASCs from all of the stages of mammary gland development (McCready et al 2014).  Here, we are focused on the question of how ASC-L and tumor cells are interacting to promote macrophages migration into the tumors.  We have clarified the rationale for this experiment (line 207).

  1. Fig. 2 (e): Authors claim that there is increase proliferation of breast cancer cells in conditioned media co-cultured from macrophage+ASC-Ls. There are huge error bars (are these Standard deviations)? It is hard to interpret significance level for individual conditioned media vis-a-vis co-culture. What is being quantified is not clear here? Is this assay similar to colony formation assay? Please clarify. Authors should show a better representative image for Fig. 2 (d). There is no scale bar. 

We agree that there was variability in this experiment.  This experiment was conducted as a traditional proliferation assay: counting 4T1-12B cells with a hemocytometer at various timepoints.  The 4T1-12B cells were treated with conditioned media collected from macrophages, ASC-L, or macrophages and ASC-L cultured together.  This description has been clarified in the methods section (lines 151-153).  We have also revised the results for this section:  there is no significant difference in the individual conditioned media versus the co-culture.  The significance arises when one compares cells receiving conditioned media versus serum-free media.   We have clarified this description of our results in lines 231-234.  We have also added a scale bar to the representative image in figure 2d as suggested.

  1. Conditioned media has various factors and in general known to promote proliferation. How can we attribute this effect to specifically macrophage proliferation? There should be some control to make this claim.

We agree that there are multiple factors that cause proliferation.  In Figure 2, we have collected conditioned media from cultured macrophages, ASC-L, or the combination of these cells together, and we tested the ability of the conditioned media to enhance tumor cell proliferation.  In figure 4, we further investigate specific cytokines produced by macrophages and ASC-L that may contribute to tumor cell proliferation and potentially angiogenesis. 

  1. Fig 3 (b): Increased cellular proliferation should be cross validated using FACS. Fig. 3 (c) To measure HUVEC cells migration why authors preferred wound healing assay over transwell migration? Any specific reason? Is 6 hr enough for HUVEC cells for wound closure ?

In Figure. 3, we have used 3 different methods to test the angiogenic potential of the secretion of ASC-L and macrophages.  While we agree that flow cytometry could be used to measure proliferation, proliferation is one measure of angiogenesis.  We have previously used wound healing assays to assess the migration of endothelial cells.  We have added images to Figure 3 of both the initial timepoint and the 6-hour timepoint to improve visualization of the wound closure.       

Round 2

Reviewer 2 Report

1. Though the authors provided the images of endothelial cell tube formation with different cultural conditions, quality of the images was poor and images with better clarity will be helpful for the readers.

2. Fig. 2d and Fig 3a are showing the co-culture of Macrophages  and ASC-L. Though the images in Fig 3a is revealing the tube formation, no such morphology is seen in Fig 2d. Is it a time dependent phenotype ? If so, at what time point was the co-culture displaying such morphology, post culturing?

Precising lengthy sentences will help readers for better understanding

Author Response

We thank the reviewer for the comments.  We have edited the revised manuscript and shortened some of the longer sentences in the introduction and discussion sections. 

  1. Though the authors provided the images of endothelial cell tube formation with different cultural conditions, quality of the images was poor and images with better clarity will be helpful for the readers.

We have replaced the jpg images with tiff images in Figure 3a.

  1. 2d and Fig 3a are showing the co-culture of Macrophages  and ASC-L. Though the images in Fig 3a is revealing the tube formation, no such morphology is seen in Fig 2d. Is it a time dependent phenotype ? If so, at what time point was the co-culture displaying such morphology, post culturing?

Figure 2d is a representative image of the cells in co-culture.  Figure 3a shows representative images of the wound healing assay.  We performed this assay using HUVEC cells.  The labeling ‘co-culture’ in Figure 3a indicates that the conditioned media applied to the HUVEC cells was isolated from macrophages and ASC-Ls cultured together, not that the image is from co-cultured cells.  We have revised the figure legend to address this confusion.